# Occupancy of the HbYX hydrophobic pocket is sufficient to induce gate opening in the archaeal 20S proteasomes

Janelle JY Chuah[1], Madalena R Daugherty[1], David M Smith[1,2]*

[1]Department of Biochemistry and Molecular Medicine, West Virginia University School of Medicine, Morgantown, United States; [2]Department of Neuroscience, Rockefeller Neuroscience Institute, West Virginia University, Morgantown, United States

## eLife Assessment

This **valuable** manuscript describes cryo-EM structures of archaeal proteasomes that reveal insights into how occupancy of binding pockets on the 20S proteasome regulates proteasome gating. The evidence supporting these claims is **convincing**, although the extrapolation of these findings to the more complex eukaryotic proteasome may prove challenging. This work will be of high interest to researchers interested in proteasome structure and regulation.

*For correspondence: dmsmith@hsc.wvu.edu

**Abstract** Enhancing proteasome function has been a long-standing but challenging target of interest for the potential treatment of neurodegenerative diseases, emphasizing the importance of understanding proteasome activation mechanisms. Most proteasome activator complexes use the C-terminal HbYX (hydrophobic-tyrosine-almost any residue) motif to bind and trigger gate-opening in the 20 S proteasome. This study defines a critical molecular interaction in the HbYX mechanism that triggers gate opening. We focus on the Hb site interaction and find it plays a surprisingly central and crucial role in driving the allosteric conformational changes that induce gate opening in the archaeal 20 S. We examined the cryo-EM structure of two mutant archaeal proteasomes, αV24Y T20S and αV24F T20S. These two mutants were engineered to place a bulky aromatic residue in the HbYX hydrophobic pocket; both mutants are highly active, though their mechanisms of activation are undefined. Collectively, our findings indicate that the interaction between the Hb group of the HbYX motif and its corresponding hydrophobic pocket is sufficient to induce gate opening in a mechanistically similar way to the HbYX motif. The activation mechanism studied here involves the expansion of the hydrophobic binding site, allosterically altering the state of the IT switch, thus triggering gate opening. Furthermore, we show that the canonical αK66 residue, previously understood to be critical for proteasome activator binding, also plays a key role in stabilizing the open gate, irrespective of activator binding. This study differentiates between the residues in the HbYX motif that support binding interactions ('YX') versus those that allosterically contribute to gate opening ('Hb'). The insights reported here will guide future drug development efforts, particularly in designing small molecule proteasome activators, by targeting the identified hydrophobic pocket.

## Introduction

The proteasome is a molecular machine responsible for degrading misfolded, damaged, and unneeded proteins that are targeted via multiple mechanisms, including the conventional pathway of ubiquitination (*Cohen-Kaplan et al., 2016*; *Bialek et al., 2023*; *Demasi and da Cunha, 2018*; *Khaminets et al.,*

*2016*; *Makaros et al., 2023*; *Tsuchiya et al., 2020*). Proteasomal degradation regulates major cellular processes and functions, such as synaptic plasticity and memory formation (*Hegde, 2017*; *Park and Kaang, 2019*; *Patrick et al., 2023*; *Lip et al., 2017*; *Jarome and Devulapalli, 2018*). Impairment of proteasome function has correlated with areas of the brain affected by neurodegenerative diseases, supporting its implication in these diseases (*Keller et al., 2000*; *Ciechanover and Brundin, 2003*; *McNaught et al., 2001*; *Ortega et al., 2007*; *McKinnon and Tabrizi, 2014*; *McNaught et al., 2002*). Furthermore, increasing proteasome activity has been shown to induce protective or rescue effects in models of neurodegenerative diseases (*Anderson et al., 2022*; *Njomen and Tepe, 2019*; *VerPlank et al., 2019*; *VerPlank et al., 2020*; *Myeku et al., 2012*; *Wang et al., 2010*), strengthening the rationale to develop small molecule proteasome activators. The objective of this study is to provide a clear mechanistic framework of how proteasome activation occurs to inform drug discovery efforts.

The eukaryotic proteasome, also referred to as the 20 S or the Core Particle (CP), consists of four stacked heteroheptameric (homoheptameric in archaea) rings arranged as α – β – β – α (*Groll et al., 2000*). Three of the seven β subunits carry protease sites with distinct protease activities: caspase-like (β1), trypsin-like (β2), and chymotrypsin-like (β5). The α subunits have N-termini that extend toward the central pore, forming a gate that regulates substrate entry. The gate is primarily formed by the N-termini of α2, 3, and 4 with α3's N-termini (the longest) acting as the lynchpin (*Groll et al., 2000*). The interactions that stabilize the open conformation have been reported in crystal and cryo-EM structures (*Groll et al., 2000*; *Rabl et al., 2008*; *Ding et al., 2019*; *Dong et al., 2019*; *Wehmer et al., 2017*; *Eisele et al., 2018*; *Förster et al., 2005*; *Chuah et al., 2023a*). A YDR motif, found on the α N-termini, interacts with the neighboring α subunit's N-terminal tail to stabilize the gate in the open state. Most recently, we discovered that the open and closed states of the α N-termini conformation are stabilized by two alternating conformations of the IT switch (*Chuah et al., 2023a*). Two α subunit residues, I12 and T13, in the archaeal *Thermoplasma acidophilum* proteasome (T20S), switch positions to occupy the pocket adjacent to Helix 0 and stabilize the open and closed gate conformation, respectively. This IT switch is functionally conserved in the eukaryotic 20 S (*Chuah et al., 2023a*). Basal kinetics of the archaeal proteasome gate, quantified using NMR (*Religa et al., 2010*), indicates that the gate fluctuates between open and close on a time scale of seconds; however, the gate primarily exists in the closed state of isolated 20 S.

The 20 S proteasome is known to associate with regulatory complexes, collectively referred to as Proteasome Activators (PAs) (*Stadtmueller and Hill, 2011*). PA700 or 19 S, which associates with the 20 S to form the 26 S, has been identified as a critical component of the Ubiquitin-Proteasome Pathway. PA700 recognizes ubiquitinated substrates and unfolds them to be degraded by the 20 S. Like many PAs, PA700 induces 20 S gate-opening via the HbYX-dependent mechanism, as does the archaeal homolog of PA700 called PAN (*Stadtmueller and Hill, 2011*; *Smith et al., 2007*). Three of the six C-terminal tails of PA700's ATPases carry a HbYX motif (hydrophobic-tyrosine-almost any residue), whereby the 'X' residue must be the C-terminal residue. These HbYX motifs interact with the inter-subunit pockets in the 20 S α ring and allosterically induce gate opening. The precise molecular mechanism of how the motif's binding in inter-subunit pockets induces gate opening is not known, though high-resolution structures of the 26 S (*Ding et al., 2019*; *Dong et al., 2019*; *Wehmer et al., 2017*; *Eisele et al., 2018*; *de la Peña et al., 2018*) and the 20 S interacting with a di-peptide HbYX-mimetic *Chuah et al., 2023a* have generated plausible models. Structures of the 26 S reveal that the C-terminal tails bind in a particular order during activation, though the correlation between the number of C-terminal tails/HbYX-bound to induce gate opening is controversial. A study indicated that partial gate opening occurs when Rpt2, 3, and 5, which have HbYX motifs, dock in their corresponding α inter-subunit pockets, and full gate opening occurs when Rpt6, which does not have a HbYX motif, additionally binds (*Ding et al., 2019*). Another study indicated that the binding of Rpt2, 3, and 5 did not open the gate, but the additional binding of Rpt6 causes full gate opening (*Wehmer et al., 2017*). Interestingly, multiple studies suggest that Rpt1, which carries the penultimate tyrosine, but not the Hb group (partial-HbYX), must also bind in combination with Rpt2, 3, 5, and 6 for gate opening (*Dong et al., 2019*; *Eisele et al., 2018*; *de la Peña et al., 2018*). Collectively, these structures suggest that each inter-subunit pocket contributes to gate opening to varying extents, which correlates with the fact that each α subunit's N-terminal tails differentially participate in the closed state, though the minimum occupancy of inter-subunit pockets required for gate opening is unresolved.

While some PAs are HbYX-dependent like PA700, the 11 S family, such as PA26 or PA28 activate the proteasome via a HbYX-independent mechanism. Despite the divergence between mechanisms, they converge on the critical molecular interactions and conformational changes (e.g. the IT switch) that stabilize the open gate state (*Chuah et al., 2023a*). For example, both HbYX-dependent and HbYX-independent PAs have been shown to interact with αK66 side chain and mutating αK66 to alanine impedes PAs association and activation, validated by velocity sedimentation assay and proteasomal peptidase activity (*Förster et al., 2005*; *Smith et al., 2007*). Both mechanisms also involve a displacement of αPro17 (*Förster et al., 2005*; *Chuah et al., 2023a*) and switching of the IT switch (*Chuah et al., 2023a*). Therefore, though the cause of the conformational changes mentioned may differ between HbYX-dependent and HbYX-independent mechanisms, they share some of the PA-20S interactions that support the open conformational state of the gate.

Our early studies showed that the HbYX motif could function even on short 6–7 residue peptides (*Smith et al., 2007*) recently, we showed that HbYX-dependent gate opening could be reduced to a carboxybenzyl (CBZ)-blocked YA dipeptide called ZYA—a minimal HbYX mimetic. Our ZYA-T20S structure (*Chuah et al., 2023a*) indicated the minimum molecular interactions necessary for optimal HbYX-dependent gate-opening by a small molecule. In our 1.9Å resolution structure, ZYA was observed to interact with residues in the α subunit (G19, K66, L81, etc.) in a similar orientation and fashion as the native HbYX motif of various PA complexes. Our cryo-EM data indicated that αV24 (and αA154) hydrophobically interacts with the carboxybenzyl of ZYA and the hydrophobic side chain of the PAs HbYX motif. Interestingly, our previous study also indicated that a single residue mutation on αV24 (i.e. αV24Y, αV24F) increased proteasome activity, though the mechanism of activation by this mutation was not determined. To assess the significance of these mutations in activating the proteasome via the HbYX-dependent pathway, we determined the cryo-EM structure of αV24Y T20S and αV24F T20S. The results of this study indicate that occupancy of a hydrophobic pocket by a bulky hydrophobic group such as αV24Y within the α inter-subunit pocket is sufficient to trigger 20 S gate opening via the HbYX-dependent mechanism, demonstrating the surprising importance of the HbYX motif's single Hb group for allosteric activation of the 20 S. The findings reported here are expected to translate to the human proteasome as ZYA is a robust activator of the mammalian 20 S and the HbYX-dependent mechanism is conserved; thereby, uncovering a highly specific pharmaceutical target for proteasome activation.

## Results

### Cryo-EM map of αV24Y T20S shows HbYX-dependent-like conformational changes and an open gate state

To elucidate how the mutation, αV24Y, induces gate-opening in the archaeal proteasome (αV24Y T20S), we resolved its structure using cryo-EM (*Figure 1*; *Figure 1—figure supplements 1–2*). The *Thermoplasma acidophilum* 20 S (T20S) has D7 symmetry, which was applied during reconstruction, and the final map resolved to 2.38 Å (EMD-44914). Comparing the αV24Y T20S electron density map against our wild-type T20S (WT T20S) map (EMD-28878) revealed conformational changes similar to those previously noted in our ZYA-T20S model (*Chuah et al., 2023a*) (PDB:8F7K) (*Figure 1*). Alignment of the maps indicate the αV24Y mutation induced minimal conformational changes in the β rings, while the α subunits exhibited rotational changes that closely mimicked HbYX-dependent conformational changes, such as those induced by ZYA binding (*Chuah et al., 2023a*; *Yu et al., 2010*; *Figure 1A & B*). Comparison of the maps show the presence of the expected larger density of the mutant tyrosine side chain in the hydrophobic pocket of the αV24Y T20S map with proximity to the docking location of the carboxybenzyl group (Z) of ZYA (*Figure 1C, D & E*).

Our prior biochemical analysis showed that the αV24Y mutation caused substantial acceleration of peptide degradation, suggesting that this mutation might stimulate gate-opening *Chuah et al., 2023a*; therefore, we sought to determine the conformational changes in the structure of αV24Y T20S. We observed significant densities corresponding to the YDR motif in the open state, designating an open gate conformation (*Figure 1F and G, & H*). Additionally, the densities corresponding to the 'IT switch' were in similar confirmation to previously known open gate proteasome cryo-EM structures: ZYA-T20S (*Figure 1I and J*, & K) and L81Y-T20S (*Chuah et al., 2023a*). It is also apparent in the map of αV24Y T20S that there is a lack of densities in the central cavity of the α ring that corresponds to

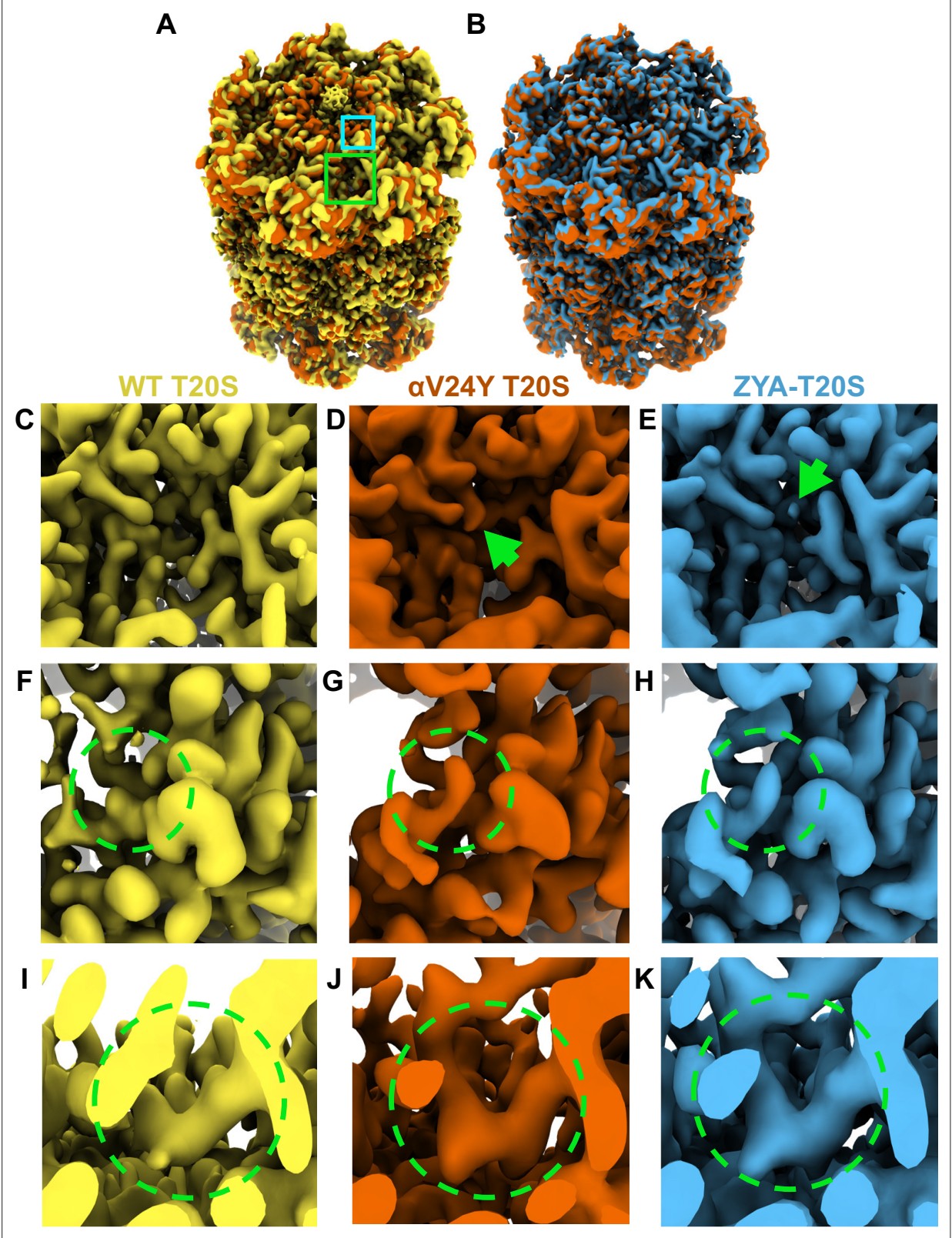

**Figure 1.** Overlay of cryo-EM maps shows αV24Y T20S has similar conformational changes as ZYA bound T20S and is in the gate-open state. (**A**) Overlay of wild-type (WT) T20S (yellow) and αV24Y T20S (red) electron density maps, showing the α and β rings. Green box outlines an intersubunit pocket and marks the zoomed region within C-E. Neon Blue box outlines the YDR motif and marks the zoomed region shown in F-H. Whole maps were aligned to one another. (**B**) Overlay of αV24Y T20S (red) and ZYA-T20S (blue) electron density maps, showing the α and β rings. V24Y T20S shows similar but larger

*Figure 1 continued on next page*

*Figure 1 continued*

conformational changes than does ZYA-T20S. (**C**) Close up of the intersubunit pocket in the WT T20S (yellow) electron density map, highlighted in (**A**) in the larger neon green box outline. (**D, E**) Same as C, but for αV24Y T20S (red) and ZYA-T20S (blue). Arrows point to the density corresponding to αV24Y and Z of ZYA, respectively. (**F**) Close up of the WT T20S (yellow) electron density map, corresponding to the YDR motif, the missing density for tyrosine is highlighted by the neon green dotted circle, which is expected in the closed state. The relative position of the close up is indicated by the smaller neon blue box outline in A. (**G, H**) Same as F, but for αV24Y T20S (red) and ZYA-T20S (blue). Neon green dotted circle highlights the YDR tyrosine side chain density, which is expected to be in this position in the open gate conformation. (**I**) Close up of the WT T20S (yellow) electron density map, corresponding to the IT Switch, highlighted by the neon green dotted circle. (**J, K**) Same as (**I**) but for αV24Y T20S (red) and ZYA-T20S (blue). Changes in the conformation of the IT switch indicates switching from the closed to the open state.

The online version of this article includes the following figure supplement(s) for figure 1:

**Figure supplement 1.** αV24Y T20S Processing Scheme.

**Figure supplement 2.** αV24Y T20S Validation.

the closed state of the α N-termini, definitively confirming that V24Y T20S's gate is open (*Figure 1A & B*). Collectively, the comparison of maps demonstrates that V24Y T20S is in an open state with global conformational changes in the α-subunits that mirror those of a HbYX-bound proteasome (*Figure 1*).

## αV24Y T20S shows HbYX-dependent conformational changes that are greater in magnitude than those seen in the ZYA-T20S structure

To gain a more detailed understanding of the conformational changes induced by αV24Y, we generated an atomic model using the αV24Y T20S map. To evaluate the global conformational changes in the α subunits, we aligned the αV24Y T20S and the ZYA-T20S to the WT T20S model via their β subunits and overlayed all three models (*Figure 2*). The α subunits in αV24Y T20S rotate roughly as a rigid body on a pivot around Helix 2 (*Figures 2 and 3A*), using WT T20S as a reference. This rotation around Helix 2 is also true for the ZYA-T20S structure and was noted previously (*Chuah et al., 2023a*), but the rotation in αV24Y T20S is greater than in ZYA-T20S (see arrows in *Figure 2A*). The helical bundle composed of Helices 4, 5, and 6 (see asterisks in *Figure 2A*) move the most with some residues in these helices moving 3–4 Å in the αV24Y T20S model relative to WT, which is about twice the distance moved in the ZYA-T20S model (*Chuah et al., 2023a*). In addition, αP17 in the αV24Y moves about 1.6 Å and Helix 0 moves ~1.4 Å compared to WT (*Figure 2C*), which is also about twice the distance these residues move in the ZYA-T20S structure. Taken together, the conformational changes induced by αV24Y are highly similar to those induced by binding of the HbYX mimetic ZYA, but they are of greater magnitude in these structures. The observed greater change is likely attributable to the αV24Y mutation affecting the entire proteasome population within that structure. In contrast, ZYA is a low-affinity ligand, anticipated to occupy only a subset of the binding sites in its structure's population. Since cryo-EM analysis averages all structures, the reduced population with conformational changes in the ZYA-bound structure represents an average of closed and open states and thus, apparent conformational changes are smaller in the ZYA structure relative to the αV24Y mutant.

αK66 has long been known to be critical for binding to the C-terminal carboxyl of PA's (*Förster et al., 2005*; *Chuah et al., 2023a*; *Smith et al., 2007*). We previously observed that αK66 was repositioned by the binding of ZYA's carboxyl (*Chuah et al., 2023a*). Since the inter-subunit pocket in the αV24Y T20S has no ligand bound, we expected αK66 to have a similar conformation to WT, but we were surprised to find that K66 was similarly repositioned in the αV24Y T20S mutant, as it was in ZYA-T20S, except to a slightly greater extent (*Figure 2D*). This suggests αK66, by itself, may play mechanistic roles in gate opening, in addition to its role in PAs binding. Based on the comparison of the WT, ZYA bound and αV24Y T20S structures, we conclude that the αV24Y mutation by itself induces HbYX-dependent conformational changes in the 20 S proteasome.

## Comparison of V24Y T20S to the WT T20S reveals specific intra-subunit conformational changes induced by αV24Y mutation

It is remarkable that a single point mutation in the T20S can mimic a 3-residue motif, let alone the activation capacity of a large complex like PAN or PA26. While the α ring is an allosteric system with each neighboring α subunit affecting the next (e.g. a rigid body rotation of one α subunits is expected to affect its neighbor and so on), it's also likely that HbYX binding or the αV24Y mutation could cause

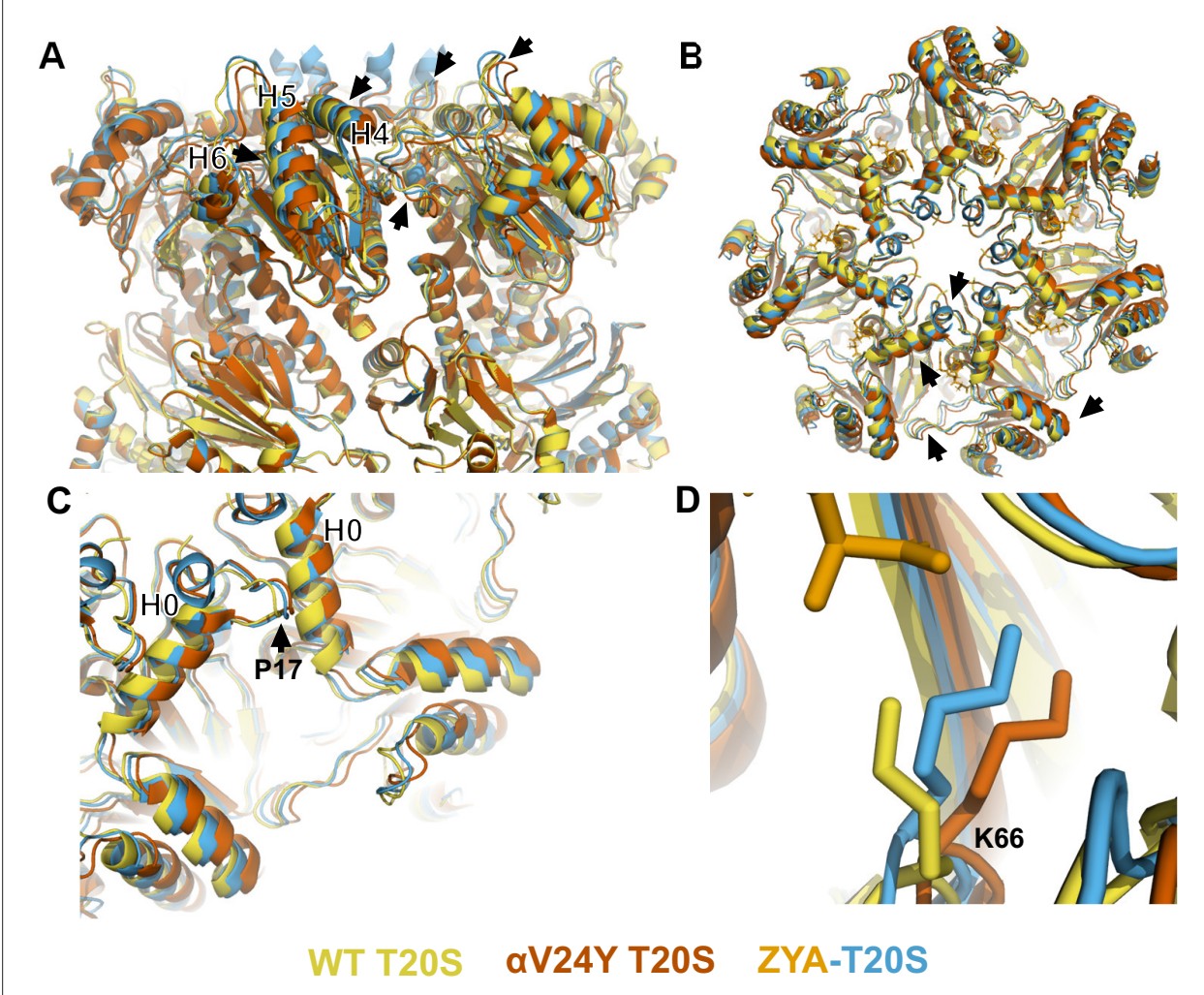

**WT T20S** **αV24Y T20S** **ZYA-T20S**

**Figure 2.** Atomic model of αV24Y T20S compared to wild-type (WT) and ZYA-T20S shows it has undergone HbYX-dependent conformational changes associated with 20 S gate-opening. (**A**) Atomic model overlay of WT T20S (yellow), αV24Y T20S (red), and ZYA-T20S (blue) with bound ZYA (gold), aligned by the β rings to show conformational changes in the α subunits. Top half of sideview of 20 S is shown. Conformational changes in the α-ring are clearly visible (black arrows). Helix 4, 5, and 6 are labeled with black asterisks outlined in white. (**B**) Top view of A. Conformational changes noted by (black arrows). (**C**) Close up view of one of the α intersubunit pockets as in B. αPro17 is shown with a black arrow. Helix 0 is labeled with a black asterisk outlined in white. (**D**) Close up view of αK66 which is labeled and shown in sticks. Models in panel b through d are aligned to show intersubunit conformational changes (i.e. aligned by β rings).

conformational changes in a single α subunit as well, which could trigger allosteric transitions to the open state. Thus, we sought to identify the intra-subunit (within a single subunit) conformational changes that could lead to larger inter-subunit (between subunits) conformational changes (*Figure 3A & B*) that perpetuate around the entire α-ring and induce gate-opening (*Figure 2B*).

The substitution of αV24 for tyrosine presents a much larger and aromatic side chain in the hydrophobic pocket. This tyrosine could impose intra-subunit conformational changes on the surrounding secondary structures. To measure these changes, we aligned a single α subunit from the WT T20S to αV24Y T20S. We found that the distance between αL21 and αA154, residues that surround the hydrophobic pocket, indicates that the pocket is widened by ~1.3 Å due to the αV24Y mutation (*Figure 3C* vs. D). Another measurement of this pocket width between T124 and L21 also widens by ~1 Å (*Figure 4G*). However, the distance between αV24 or αV24Y to αL81 is consistent (*Figure 3C vs. D*), suggesting that the hydrophobic pocket has not enlarged in all directions.

We hypothesize that the widening of the hydrophobic pocket that αV24Y occupies is the causal point of origin triggering the inter-subunit conformational changes that we observed in *Figure 2*. The

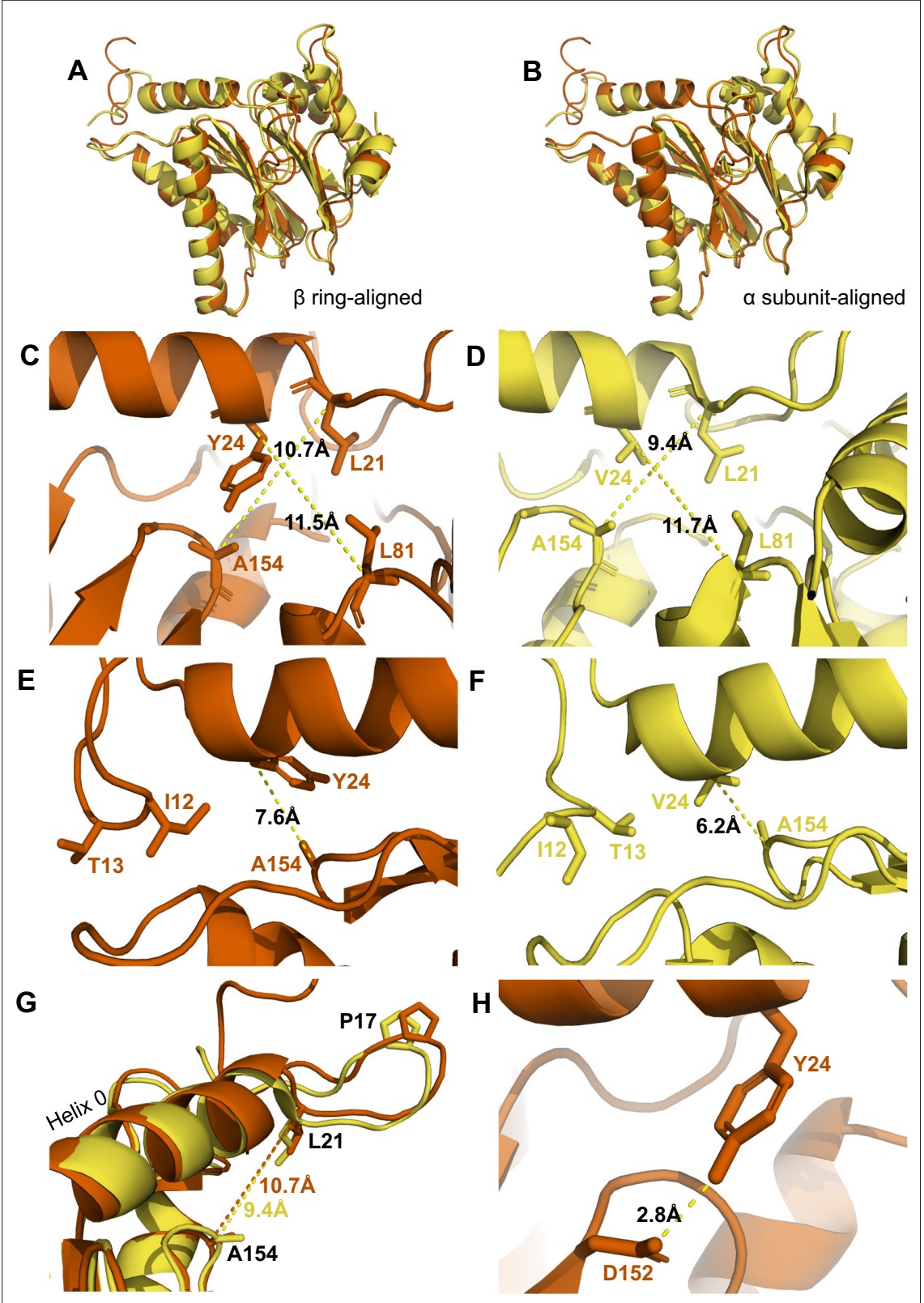

**Figure 3.** αV24Y mutation induces intrasubunit conformational changes in the HbYX hydrophobic pocket, IT switch pocket, and P17 loop. (**A**) Overlay of wild-type (WT) T20S (yellow) and αV24Y T20S (red) α subunit models, aligned by the β rings. (**B**) Same as A except the models were aligned according to individual α subunit. (**C**) Close up of the intersubunit pocket in αV24Y T20S (red), showing distances between the α-carbon selected residues (labeled and shown in sticks) to demonstrate conformational changes when compared to the WT T20S (yellow) model in D. Models are aligned by individual

*Figure 3 continued on next page*

*Figure 3 continued*

α subunit. (**D**) Same as C except it is the WT T20S (yellow) model. (**E**) Close up on the residues of the IT Switch (sticks) and αV24Y (sticks), showing the distance between the α-carbon of αV24Y and αA154 (sticks) to demonstrate conformational changes in the open state, when compared to the WT T20S (yellow) model in F. Models are aligned by individual α subunit. (**F**) Same as E except it is the WT T20S (yellow) model. (**G**) Close up of Helix 0 and selected surrounding residues (shown in sticks) in an overlay of WT T20S (yellow) and αV24Y T20S (red) α subunits, aligned by individual α subunit. Distances are measured between α-carbons and colored according to the model it corresponds to. (**H**) Close up of αV24Y (shown in sticks) in the αV24Y T20S model, demonstrating its side chain's polar interaction (yellow dotted line) with αD152 (shown in sticks).

The online version of this article includes the following figure supplement(s) for figure 3:

**Figure supplement 1.** Additional measurements in the intersubunit pocket in V24Y T20S.

**Figure supplement 2.** Conservation of T20S residues with human 20 S α subunits.

HbYX hydrophobic pocket occupied by αV24Y side chain is adjacent to another hydrophobic pocket found on the opposite side of Helix 0 (on the 20 S central channel), where the side chain of αI12 (an IT Switch residue) occupies in the open state (***Figure 3E & F***). The αV24Y T20S model coincides with the αV24Y T20S map, showing that the IT switch is in the open state (***Figure 1***), where the αI12 side chain is oriented to occupy the IT switch hydrophobic pocket adjacent to Helix 0 (***Figure 3E vs. F***). Due to the adjacent proximity of both hydrophobic pockets (on either side of and below Helix 0), a conformational change in the HbYX hydrophobic pocket would also affect the IT switch's hydrophobic pocket since these pockets share common space. It is highly plausible that the change in the HbYX hydrophobic pocket, resulting from the larger side chain of tyrosine (compared to valine), could facilitate the switch between αT13 and αI12 in the IT switch hydrophobic pocket (***Figure 3E vs. F***). The αV24Y mutation increases the hydrophobicity and size of the pocket, likely enhancing the binding affinity for the larger hydrophobic side chain of isoleucine. In conclusion, it appears that occupancy of the HbYX hydrophobic pocket with an aromatic ring of tyrosine allosterically affects the IT switch hydrophobic pocket, as intra-subunit conformational changes allow the IT switch to reconfigure to an open state.

αProline17 (P17) displacement is one of the first identified inter-subunit conformational changes associated with triggering gate opening, specifically by the 11 S family members (***Förster et al., 2005***; ***Stadtmueller and Hill, 2011***; ***Hill et al., 2002***). Interestingly, when the IT switch switches to an open configuration, it generates 'slack' between the IT switch and P17 (***Chuah et al., 2023a***; ***Figure 3—figure supplement 1A & B***). In αV24Y T20S, this 'slack' appears to allow for an increase in the intra-subunit distance between Helix 0 and αP17 (***Figure 3—figure supplement 1B***). In addition, we observe an intra-subunit displacement (not rotation-based) of P17 by 1.5 Å (***Figure 3G***) consistent with the additional slack from the switching of the IT switch. In addition, intra-subunit analysis shows αP17 is displaced by 1 Å away from its neighbor's Helix 0 (***Figure 3—figure supplement 1C***); however, this distance between P17 and the neighboring Helix 0 does not change globally (***Figure 3—figure supplement 1A & B***) due to the rigid body rotation of the α subunits which maintain packing between the P17 loop and the neighboring Helix 0. In short, the conformational change of the IT switch to the open state appears to provide flexibility for the Pro17 loop to absorb, or perhaps allow, the rigid body rotation associated with HbYX-induced gate-opening. Moreover, our structure suggests the widening of the hydrophobic pocket between αL21 and αA154 may also contribute to the shift in αP17, as αL21 is connected on the other side of the αP17 loop (***Figure 3G***).

We observed that the side chain of αE25 is involved in an intrahelical hydrogen (H)-bond with αR28 in the WT structure (***Figure 4—figure supplement 2A & B***), which is displaced by the αV24Y mutation and instead H-bonds with αR20 (***Figure 4—figure supplement 2C & D***). The displacement slightly increases the distance between αE25 and αR20 (***Figure 4—figure supplement 2B vs. D***), further providing 'slack' for the αP17 movement. We previously showed the mutation of αE25 to alanine, activated gate opening and prevented PAs from being able to induce gate opening in the T20S (***Chuah et al., 2023a***). Therefore, biochemical evidence also supports an important role for this αE25 rearrangement in destabilizing the closed state and perhaps stabilizing the open state.

We also considered other mechanisms by which the αV24Y mutation might influence gate opening, such as its side chain's hydroxyl polar interactions. We find that αY24 side chain interacted with the side chain of αD152 (***Figure 3H***); however, no significant conformational changes were observed along the β-sheet which αD152 is positioned on. Thus, it is not apparent how the polar interaction between αY24 and αD152 might be associated with gate opening, other than the fact that the H-bond may help orient this tyrosine rotamer.

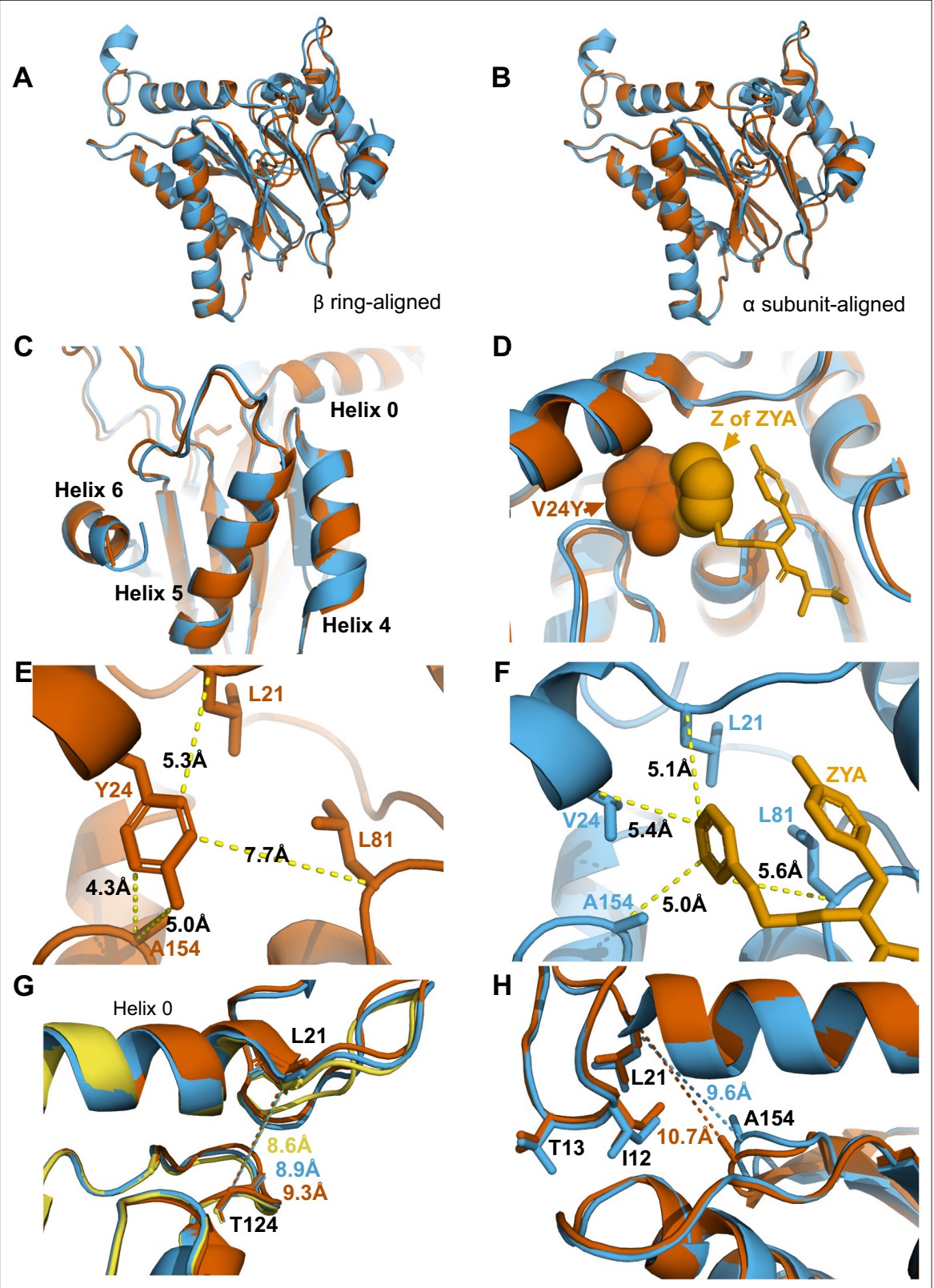

**Figure 4.** αV24Y mutation and ZYA binding both induce similar intrasubunit conformational changes. (**A**) Same as 3 A except for αV24Y T20S (red) and ZYA-T20S (blue). Models are aligned by the β ring. (**B**) Same as 3B except for αV24Y T20S (red) and ZYA-T20S (blue). Models are aligned by individual α subunit. (**C**) Same as B except showing a close up of the Helix 4, 5, and 6 bundle. (**D**) αV24Y T20S and ZYA-T20S are aligned by β subunits and αV24Y side chain and carboxybenzyl group (**Z**) of ZYA are both shown in space-fill model to appreciate the level of spatial overlap between the αV24Y tyrosine

*Figure 4 continued on next page*

*Figure 4 continued*

ring (red) and the Z ring (gold) of ZYA, demonstrating they each occupy the same HbYX hydrophobic pocket in the T20S. (**E**) Close up of the αV24Y T20S (red) α intersubunit pocket showing measurements between the αV24Y tyrosine and labeled α carbons (shown in sticks). Comparison against F indirectly demonstrates the position of αV24Y's side chain relative to the carboxybenzyl group of ZYA in the ZYA-T20S model. (**F**) Measurements of distances between ZYA's carboxybenzyl group (gold) and the α-carbon of surrounding residues (shown in sticks) in the ZYA-T20S model (blue). (**G**) The HbYX hydrophobic pocket measurement: αT124 to αL21 (main chain shown in sticks) distance in WT T20S (yellow), αV24Y T20S (red), and ZYA-T20S (blue). (**H**) Same as 3E & F, except with alignment of ZYA-T20S (blue) and αV24Y T20S (red) α subunit models. Distances noted are colored according to the models they correspond to representing intrasubunit differences.

The online version of this article includes the following figure supplement(s) for figure 4:

**Figure supplement 1.** αV24Y T20S differs from wild-type (WT) T20S.

**Figure supplement 2.** V24Y mutation and ZYA binding both alter E25's H-bonding partner.

**Figure supplement 3.** Comparison of HbYX hydrophobic pocket in V24Y T20S and ZYA-T20S structures.

**Figure supplement 4.** Comparison of surface area of HbYX hydrophobic pocket in WT T20S and V24Y T20S structures.

## Comparison of activated T20S forms: αV24Y T20S and ZYA-T20S

The cryo-EM structure presented here (*Figures 1 and 2*) and our previously published biochemical data (*Chuah et al., 2023a*) clearly demonstrate the open state of the αV24Y-T20S mutant. The substitution of αV24 for tyrosine was engineered to mimic how the 'Hb' group of the HbYX motif binds to its hydrophobic pocket, but does it? As previously mentioned, the global conformational changes in the αV24Y 20 S were similar to a HbYX-bound 20 S (i.e.minimal conformational changes in the β subunits, rigid body rotation of α subunits pivoting on Helix 2) (*Figures 1B and 2*), suggesting that the mechanism of gate-opening by αV24Y mutation has mechanistic overlap with the HbYX mechanism (e.g. ZYA-T20S). Therefore, we sought to identify, at the molecular level, how these two open T20S models compare and identify any intra-subunit conformational changes that can be specifically attributed to the occupancy of the hydrophobic pocket by the Hb component of the HbYX motif, without influence from the Y and X residues of the HbYX motif. The αV24Y T20S model allows for analysis of the isolated effects of Hb occupancy in the hydrophobic pocket.

Comparison of αV24Y T20S against ZYA-T20S reveals that the intra-subunit conformational changes induced by αV24Y were highly similar, as there were few differences between the α subunits when aligned to one another (*Figure 4A and B & C*). Comparison of the hydrophobic pocket between the two open models, ZYA-T20S and αV24Y T20S, shows that the Z group of ZYA and the αV24Y tyrosine both clearly bind to the HbYX hydrophobic pocket and overlap in space, though they bind in different orientations (*Figure 4D*). Moreover, molecular surface representations of the hydrophobic pocket (WT versus V24Y T20S) clearly show pocket occupancy by the mutant tyrosine's side chain (*Figure 4—figure supplement 4*). In addition, the measurements of αV24Y's side chain or Z of ZYA to surrounding residues within each individual subunit provide the most relevant visual approximation and overlap comparison of the 'Z' and 'αV24Y' aromatic group positions in the HbYX hydrophobic pocket (*Figure 4E & F*). For comparison, the distance between αV24 or αV24Y to αL81 marginally differs by ~0.3 Å, while the distance between αL21 and αL81 is comparable in both models (*Figure 4—figure supplement 3A & B*). The measurement of the hydrophobic pocket between αL21 and αT124 is widened in the αV24Y T20S (~0.7 Å; *Figure 4G*). This pocket measurement also enlarges in the ZYA T20S structure compared to WT, though to a lesser extent (*Figure 4G*), consistent with the observed extent of changes in these models (*Figure 2*). However, the largest difference in this comparison between ZYA bound and αV24Y is in the measurement between αL21 and αA154, as the V24Y tyrosine is better positioned to influence this specific measurement. (*Figure 4H*). To validate this important conformational change in the maps raw data in a statistically significant way, we used CCP-EM to generate confidence maps of WT and V24Y structures at an FDR of 0.01. These confidence maps clearly show conformational differences in A154 in the V24Y structure (*Figure 4—figure supplement 1B*). Lastly, the placement of αI12's side chain in the IT switch hydrophobic pocket between Helix 0 and Helix 2 protrudes ~0.8 Å deeper in the pocket in V24Y T20S compared to ZYA-T20S (*Figure 4H*), consistent with the larger hydrophobic pocket under Helix 0 in the mutant.

Collectively, evidence suggests that ZYA's mechanism as an activator is due to its hydrophobic group occupying the space between HbYX hydrophobic pocket, similar to αV24Y, but ZYA's mechanism also relies on other parts of the dipeptide as modifications to tyrosine affected the small

molecule's ability to activate (*Chuah et al., 2023b*). In combination with the evidence shown here, we conclude that the hydrophobic group of ZYA is likely the primary feature allosterically affecting gate-opening but the insertion of this Hb group into the hydrophobic pocket also relies on interactions of the Y and X residues with the inter-subunit pocket to provide sufficient binding affinity. As for αV24Y T20S, the hydrophobic component (tyrosine's side chain) is covalently held in place in the hydrophobic pocket and thus does not require the 'YX' interactions and, therefore, sufficiently supports stable gate opening through a similar mechanism to ZYA.

## αK66 supports gate opening through a previously unknown mechanism

Many studies in the past have highlighted the importance of αK66, found in the back loop of the inter-subunit pocket (*Figures 2D and 5A*), and our αV24Y T20S structure shows that αK66 is reorganized similar to when ZYA binds. These observations inspired us to elucidate how intra-subunit conformational changes in αK66 may be affected by αV24Y. αK66 is well known to form hydrogen or salt bonds with the C-termini of PAs and is required for PA binding to the 20 S (*Förster et al., 2005*), but how is αK66 affected when there is no bound PA? The back loop of the inter-subunit pocket is naturally a more flexible loop but in αV24Y T20S, this loop does have significant intra-subunit conformational changes (*Figure 5A*), with this loop being displaced by ~3 Å (inter-subunit measurements are similar). Additionally, we noticed αK66 moved ~1 Å towards αT78 and its side chain is oriented to be in closer proximity to hydrogen bond with both αT78 and αE211 (*Figure 5A & B*). Such interactions are not observed in the WT T20S (*Figure 5C*) but are seen in the ZYA-T20S (*Figure 5D*). αK66A mutants do not exhibit activation by HbYX-dependent or HbYX-independent activators leading prior studies to conclude that αK66 was important for binding of proteasome activators (*Förster et al., 2005*; *Smith et al., 2007*). However, the repositioning of αK66 in the αV24Y T20S structure, which does not have a proteasome activator bound, indicates that αK66 may also be involved in stabilizing the open state on its own. To test this hypothesis, we generated a double mutant proteasome, αV24Y-αK66A T20S. We isolated this mutant and measured fluorogenic peptide hydrolysis activity (LFP). We found that αV24Y-αK66A T20S has similar activity to WT and is not hyperactive like αV24Y T20S (*Figure 5E*). Thus, αK66 is essential for αV24Y-induced gate-opening. These enzymatic results support the conclusion that αK66 does not only interact with proteasome activators to support gate opening but is also critically involved in stabilizing the open state of the α subunit gate by forming new interactions with αT78 and αE211 in the open state .

## αV24Y mutation affected the hydrophilicity of the inter-subunit pocket

Our previous study of ZYA-T20S suggested that the HbYX mechanism of activation involved the rearrangement of water molecules, so we sought to determine if the same applies in αV24Y T20S. The mutation essentially introduces a larger hydrophobic side chain to the inter-subunit pocket and, therefore, we expect rearrangement of water molecules. Our map and model suggest that the αV24Y mutation either excluded or rearranged water molecules found in the WT T20S inter-subunit pocket (*Figure 6A vs. B*). Additionally, none of the water molecules placement in αV24Y T20S coincided with the exact placement and interactions as in ZYA-T20S (*Figure 6C*), suggesting that waters involved in ZYA's binding to the inter-subunit pocket may not be relevant in αV24Y T20S. This is unsurprising given that ZYA is a molecule that occupies a good portion of the inter-subunit pocket, relative to the small change caused by the αV24Y mutation. Interestingly, the inter-subunit pockets of αV24Y appears to be more dehydrated relative to the WT, but structures with equivalent resolutions would be necessary to confirm. The results collectively suggest that while water interactions in the inter-subunit pockets change substantially in the open versus closed states (at these resolutions), we do not observe specific waters shared between these two open 20 S models that would suggest waters being critically involved in stabilization of the open gate by these two different activation mechanisms (*Figure 6*).

## αV24F T20S is not activated similarly to αV24Y T20S

Similar to αV24Y T20S, the αV24F T20S mutant showed increased basal activity compared to WT T20S biochemically (*Chuah et al., 2023a*). Therefore, we sought to elucidate whether the mechanism of activation would be similar if the substitution was to a phenylalanine as opposed to a tyrosine. Comparison of the αV24F T20S map (*Figure 7A*, *Figure 7—figure supplements 1 and 2*) to αV24Y

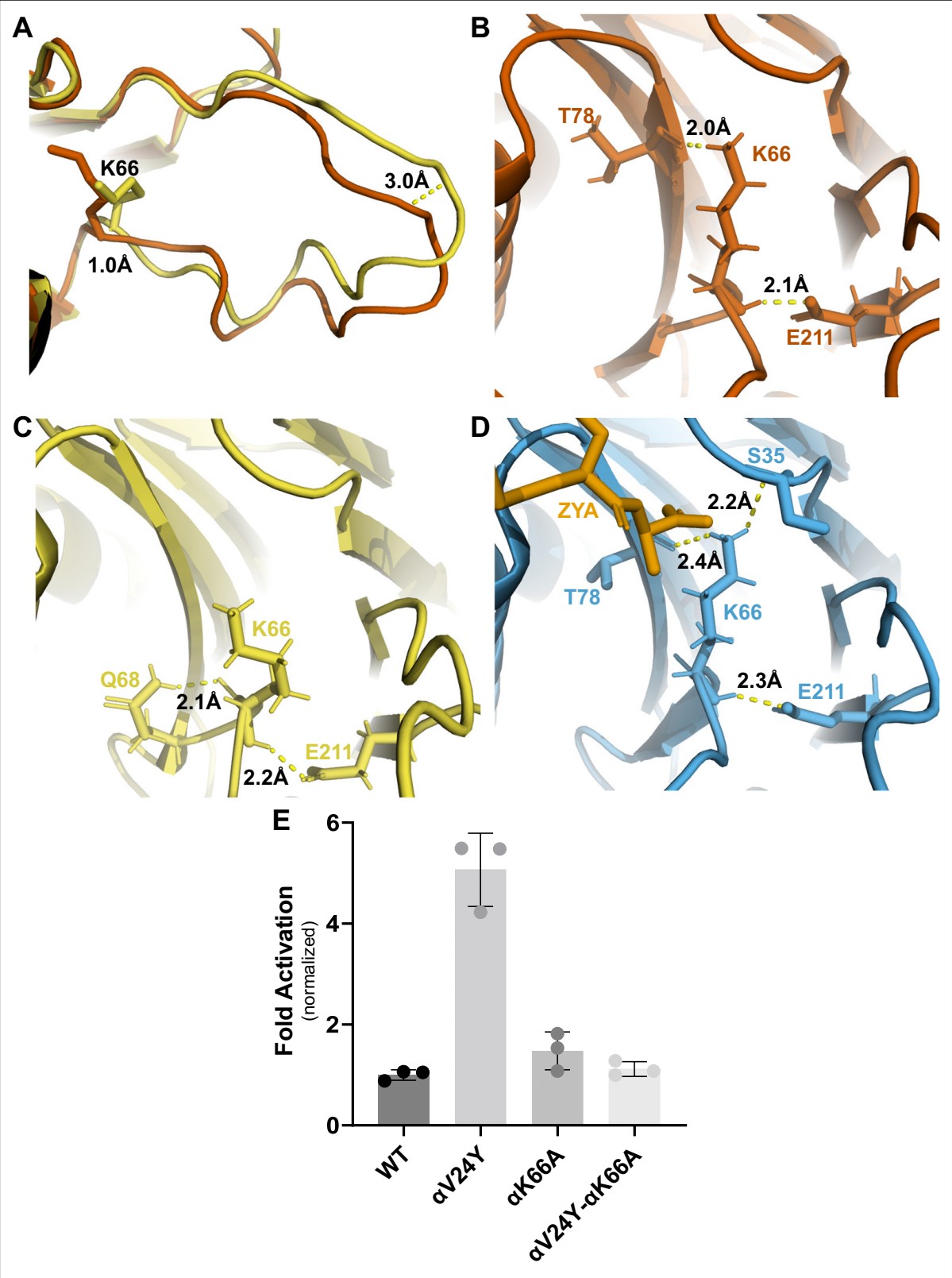

**Figure 5.** αK66 plays a key role in stabilizing the open state of the T20S, even in the absence of proteasome activator binding. (**A**) Overlay of wild-type (WT) T20S (yellow) and αV24Y T20S (red), aligned by individual α subunits, showing a close up of the αK66 (shown in sticks) back loop. Distance measured demonstrates the extent of intrasubunit conformational changes in the 'back loop' and K66 α carbon. (**B**) Close up of αV24Y T20S (red) model, showing the polar interactions (yellow dotted lines) between αK66 and surrounding residues (shown in sticks). Distances are measured between

*Figure 5 continued on next page*

*Figure 5 continued*

the exact atoms involved in the polar interactions. (**C**) Same as B except it is the WT T20S (yellow) model. (**D**) Same as B except it is the ZYA-T20S (blue) model and ZYA (gold). (**E**) WT T20S and mutants were measured for activity based on LFP degradation rate (rfu/min), normalized to the WT T20S as a control. Data (means) are representative of three or more independent experiments, each performed in triplicate. Error bars represent ± standard deviation.

T20S and WT T20S maps and models reveal a distinct lack of densities in the α subunits when αV24 is mutated to phenylalanine (*Figure 7B & C*). The lack of densities is most apparent from the N-termini and through Helix 0, though the map was also lacking densities in other selected regions of the α subunit. Globally analyzing the αV24F T20S map suggests that the αV24F mutation affected the proper folding of secondary structures surrounding the αV24F mutation (e.g. on Helix 0), but interestingly, this lack of Helix 0 folding did not affect the assembly of the proteasome itself (*Figure 7B & C*). The difference between maps indicates that the β subunits are minimally affected in the assembly of the proteasome and that the lack of densities affected the α subunit throughout, though they were most apparent around the gating regions (*Figure 7D & E*). The lack of densities in αV24F T20S map suggests that substrates could enter the mutant proteasome more easily through the larger opening and, therefore, biochemically exhibited kinetics comparable to or even exceeding the activated proteasome (*Chuah et al., 2023a*).

## Discussion

Proteasome impairment and reduced proteostasis are common observations in clinical cases and animal models of neurodegenerative diseases such as Alzheimer's and Parkinson's, which has inspired the search for small molecule proteasome activators for years. Yet, there is a lack of robust small molecule proteasome activators to date, in part due to a lack of understanding in proteasome activation mechanisms. The α inter-subunit pockets have long been known to be the binding site of PA complexes but the exact molecular interactions between the inter-subunit pockets and the PAs that cause gate-opening have been elusive, preventing efficacious design of proteasome-activating small molecules. While no FDA-approved in vivo proteasome-activating small molecule exists as a tool for proof of concept, our previous work (*Anderson et al., 2022*) to express constitutively active 20 S proteasomes in *C. elegans* indicated that the mutant proteasomes supported an increased resistance to proteotoxicities and increased lifespan, supporting the rationale for this study. Here, we present a single specific binding site within the inter-subunit pocket that when occupied, is sufficient to induce gate opening and, therefore, a plausible and well-defined pharmaceutical target for proteasome activators.

To identify the fundamental molecular interaction(s) required for gate-opening, we focused on the archaeal proteasome as it is homoheptameric, which is particularly an advantage when desiring a high-resolution cryo-EM structure to map detailed interactions. Moreover, the archaeal proteasome still carries the conserved structures involved in gating, and both human and archaeal 20 S can be opened by the same HbYX peptides (e.g. ZYA), implying that the findings of this study could be applied to the human system with some caveats discussed below. We recently demonstrated that the αV24Y mutant archaeal proteasomes exhibited upregulated activity (*Chuah et al., 2023a*). Our αV24Y T20S structure here reveals inter-subunit conformational changes essentially identical to a HbYX-bound archaeal proteasome (ZYA-T20S)(*Chuah et al., 2023a*), demonstrating that it is in an open state, allowing us to further elucidate how this single-residue mutation opens the gate, and by extension, better understand the mechanism of gate opening by the HbYX motif itself.

Collectively, our findings in αV24Y T20S focus on a mechanism of gate-opening originating at the occupancy of a hydrophobic pocket between αL21 and αT124. The αV24Y mutation places an aromatic side chain in between αL21 and αT124 (as well as αL21 and αA154), similar to the position of the benzyl group (Z) of ZYA in the ZYA-T20S structure. The occupancy of this hydrophobic pocket by αV24Y expands the adjacent pocket where αI12 of the IT Switch is previously found to bind in open gate proteasome structures. αL21, displaced by αV24Y's larger side chain compared to αV24 (*Figure 3C*), is connected to the αP17 loop, which has been commonly shown to be displaced in open gate proteasome structures. The results here also suggest that this mechanism applies to the ZYA-T20S (*Figure 4G*), though it may not be the only mechanism supporting ZYA's gate-opening as

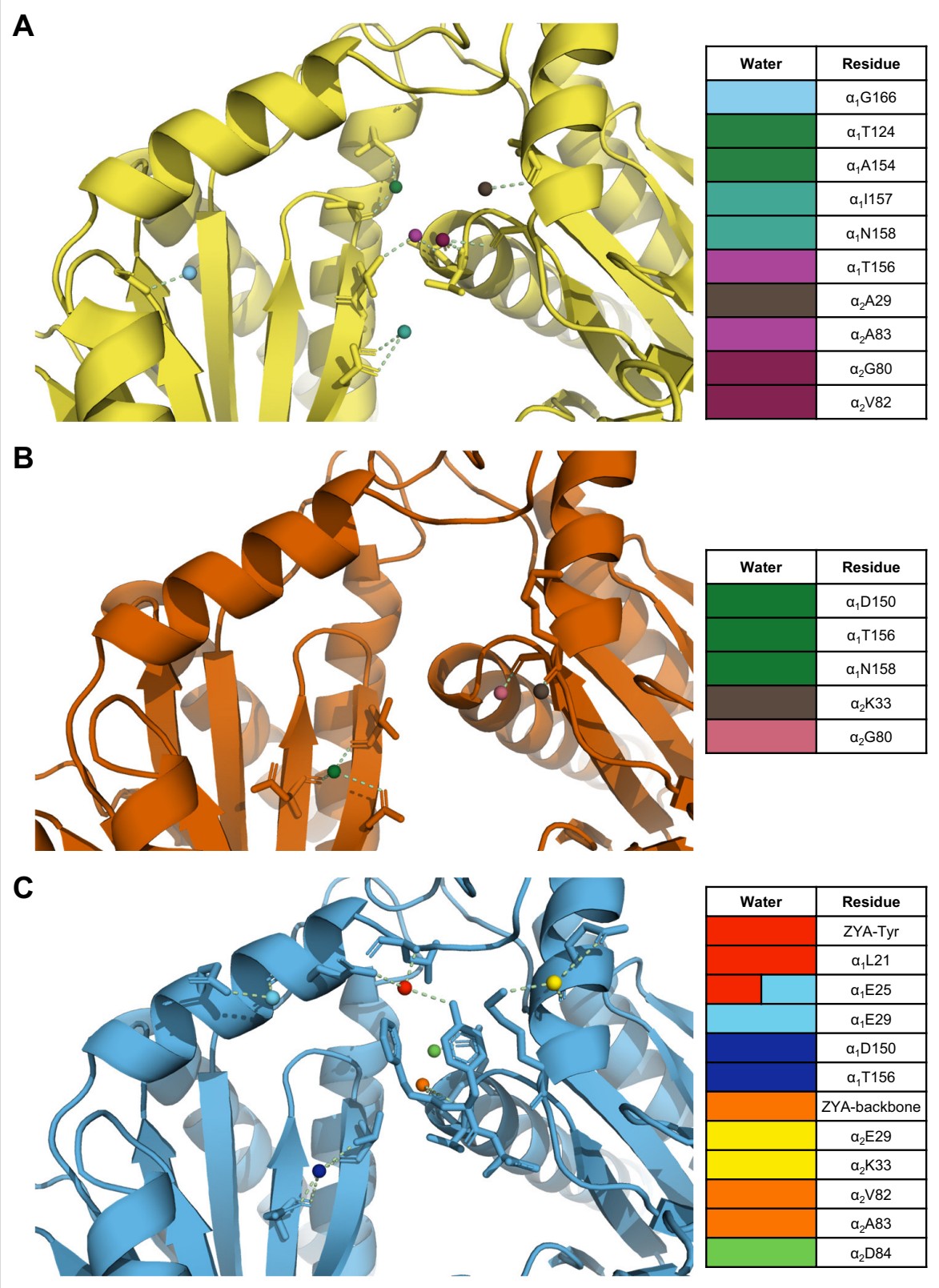

**Figure 6.** The interunit pocket of the αV24Y T20S is less populated with waters, relative to ZYA T20S and wild-type (WT) T20S. (**A**) Intersubunit pocket of WT T20S model (yellow) showing waters (spheres) and their interactions (yellow dotted lines) with the residues (shown in sticks, listed in table) in the intersubunit pocket. Spheres are colored individually and match the colors on the table (right) to identify residues it interacts with. (**B, C**) Same as A except αV24Y T20S model (red) and ZYA-T20S model (blue).

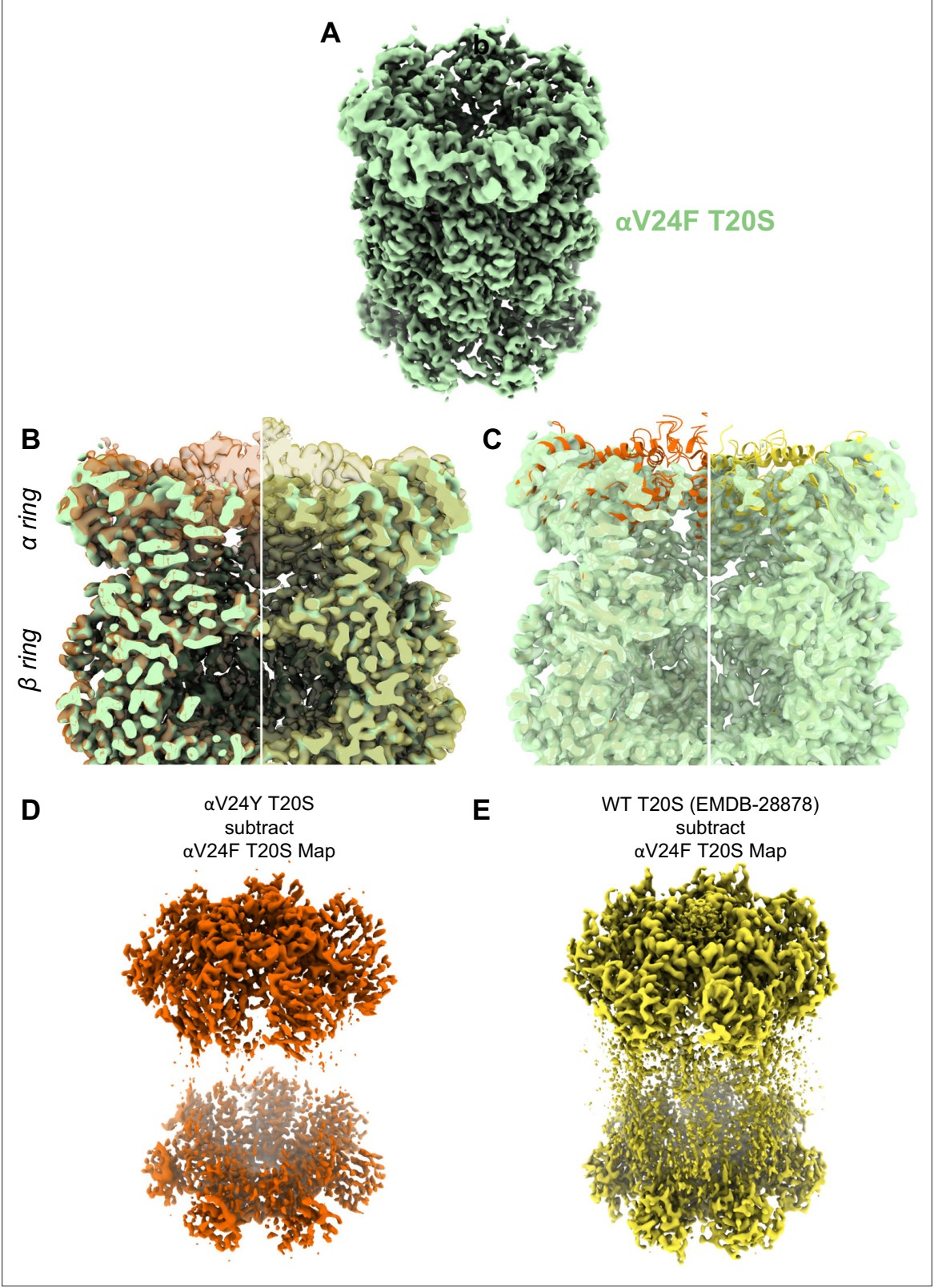

**Figure 7.** A large portion of the N-terminal domain of the αV24F T20S mutant is unstructured. (**A**) Angled view of αV24F T20S 2.2 Å electron density map (green), showing the α and β rings. (**B**) Overlay of αV24F T20S map with αV24Y T20S map (red; transparent surface; left) and wild-type (WT) T20S map (yellow; transparent surface; right). (**C**) Same as B except αV24Y T20S model (red, cartoon, left) and WT T20S model (yellow; cartoon; right).

*Figure 7 continued on next page*

*Figure 7 continued*

(**D**) Difference map generated by subtracting αV24F T20S map from αV24Y T20S map, using ChimeraX. (**E**) Same as (**D**) except WT T20S map instead of αV24Y T20S map.

The online version of this article includes the following figure supplement(s) for figure 7:

**Figure supplement 1.** αV24F T20S Processing Scheme.

**Figure supplement 2.** αV24Y T20S Validation.

ZYA is a dipeptide as opposed to a single residue mutation (αV24Y) that occupies the hydrophobic pocket. For example, the YA residues bridge the subunit from αK66 to the αP17 loop, which also likely stabilizes the open state (*Chuah et al., 2023a*). The interaction of ZYA's carboxy group with αK66 may also help stabilize the open-state conformation of αK66 (*Figure 5B*), which we showed is essential for gate-opening here (*Figure 5E*). The conformational changes between the two models are consistent, although αV24Y T20S exhibits larger conformational changes likely due to cryo-EM averaging effects in the lower populated ligand-bound states (ZYA) compared to the V24Y mutant. ZYA, as a dipeptide has the ability to and perhaps, even rely on interacting with other parts of the inter-subunit pocket to supports its binding and orient the Z group in the hydrophobic pocket. Results from both αV24Y T20S and ZYA-T20S both suggest that occupying the hydrophobic pocket is a key to gate opening in the archaeal proteasome, and potentially the mammalian proteasome if placed in the right pocket(s).

The two key residues that 'sandwiches' αV24Y's side chain and form the hydrophobic pocket, αL21 and αA154, are conserved in the human 20 S α inter-subunit pockets (*Figure 3—figure supplement 2*), indicating that a similar mechanism of occupying this hydrophobic pocket is used for gate-opening in the human system. Five out of seven H20S α subunits carry leucine in a similar position to αL21 in the T20S. The remaining two subunits carry an isoleucine (α6: PSMA1) and valine (α7: PSMA3), which are hydrophobic like leucine. Additionally, while αA154 is only conserved as alanine in α1: PSMA6, the rest of the subunits carry a methionine in this position, which is also hydrophobic. Previous analysis (*Chuah et al., 2023a*) also indicated that the H20S has analogs of the IT switch in selected α subunits, which suggests that targeting this hydrophobic pocket at selected α subunits with the conserved IT switch would have a greater effect on gate-opening than others. Any application of this mechanism in the human system is still complicated and should also consider the prior prediction that multiple inter-subunit pockets likely need to be occupied by the HbYX motifs for gate-opening (*Chuah et al., 2023b*). The conservation of activation motif, residues, and conformational changes between archaeal and human proteasomes for gate opening demonstrates potential for the findings in this study to translate to activating the human 20 S.

Interestingly, our αV24F T20S structure indicates that the position of this residue in the protein chain plays a role in α subunit folding. Our αV24F T20S cryo-EM map reveals that the α subunits are partially assembled, with the majority of regions proximal to the gating residues lacking densities, suggesting disorder. Yet, the β subunits appear functional, both structurally and biochemically (*Chuah et al., 2023a*), which suggests that the proteasome does not require perfectly formed α subunits for complete proteasome assembly. We notice that the hydroxyl group of αV24Y's side chain is involved in polar interactions; however, we do not presume this to be involved in α subunit assembly as valine lacks the side chain to interact similarly. Rather, we speculate that αV24F may introduce too large of a hydrophobic group that, without a hydroxyl to interact with αD152 (*Figure 3H*) prevents proper folding interactions.

Collectively, our analysis indicates that the HbYX hydrophobic pocket expands due to αV24Y's more bulky side chain, which likely sets of a chain of events starting with the IT switch, which are modeled and schematized in *Figure 8*. The wider and now more hydrophobic IT switch pocket likely promotes IT switch to switch to the open state, providing flexibility for the αP17 loop intra-subunit motions, or 'slack.' In addition, the widening of the hydrophobic pocket and rearrangement of the αE25 H-bonding pattern also appears to allow more flexibility for αP17 loop motions. Combined, these intra-subunit conformational changes collectively elongate the αP17 loop, increasing its flexibility. This enhanced flexibility is crucial because the αP17 loop interfaces with Helix 0 of neighboring α subunits. The loop's increased plasticity accommodates rigid-body rotations against the adjacent α subunits, which would otherwise be constrained. As a result, these rotational movements can propagate allosterically around the proteasome α ring. Though this causal chain of events is supported by

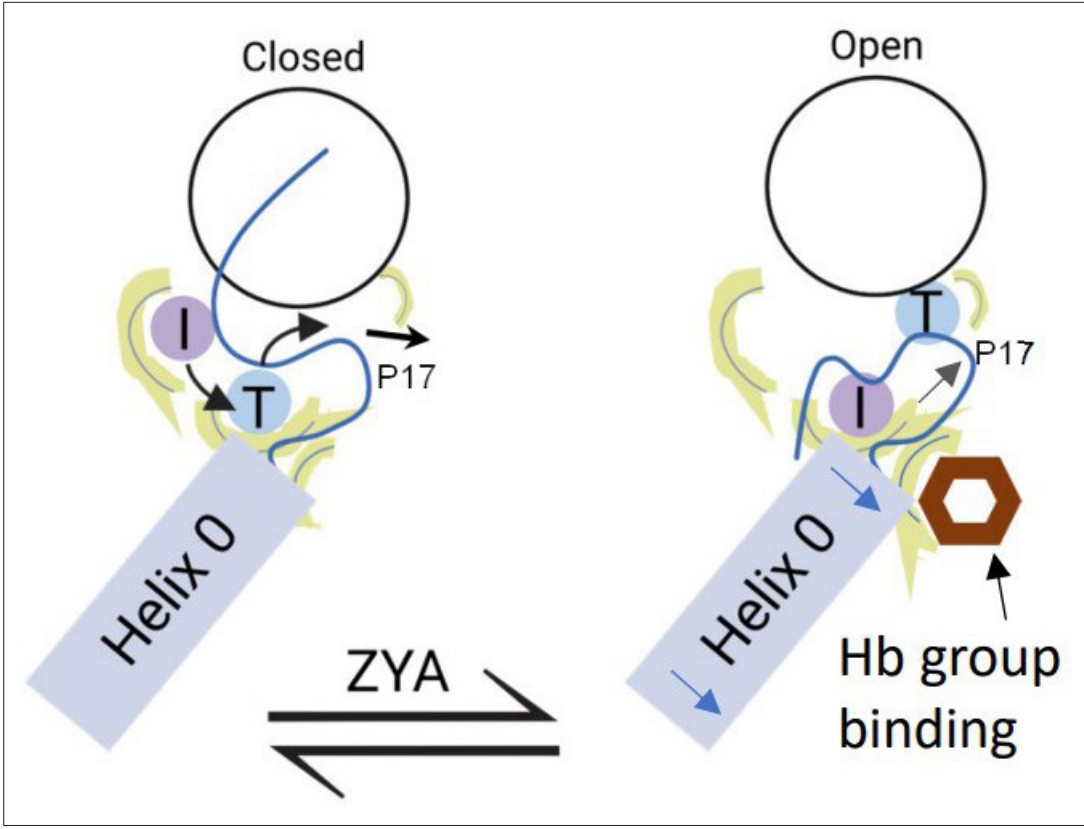

**Figure 8.** Model for induction of gate opening by binding of the Hb group binding to the HbYX hydrophobic site in the 20 S intersubunit pockets. Binding of the Hb group of the HbYX motif, or tyrosine of αV24Y, is expected to initiate an allosteric chain of events that leads to 20 S gate opening. Hb group binding causes.

these structures, the timing and order of these events cannot be elucidated with these static structures. The structures here suggest a plausible molecular mechanism of how a single residue mutation in the HbYX hydrophobic pocket can induce 20 S gate opening (*Figure 8*).

Our findings in this study also highlight and revise an understanding of the αK66's role in proteasomal gate-opening. Previous findings have consistently alluded to its critical role in the association of HbYX-dependent and HbYX-independent PAs, as αK66 was shown to interact with and orient the C-terminal tails of PAs for gate-opening (*Förster et al., 2005*; *Chuah et al., 2023a*; *de la Peña et al., 2018*; *Hill et al., 2002*). To our surprise, the αV24Y T20S structure demonstrates that αK66's side chain conformation changes, even though it does not interact with a PA or HbYX mimetic such as ZYA. Our double mutant αV24Y&K66A T20S's basal activity supports the necessity of the αK66 conformational change observed in the αV24Y model and indicates the involvement of αK66 in stabilizing the open-gate conformation through a previously unreported mechanism. The conformational change in αK66 is supported by changes in H-bonding interactions between αK66 and surrounding residues, which are absent in the closed structure (WT T20S). These findings, therefore, demonstrate that αK66's conformational changes support other conformational changes such as the conformation of the back loop, which it is connected to, and has been shown to be involved in HbYX-dependent gate opening (*Chuah et al., 2023a*). Essentially, αK66 is deduced to contribute more than interacting with activators, and also appears to be necessary to stabilize the open state of the 20 S.

Our prior study showed the possible roles waters could play in proteasome activation and gating regulation. Our ZYA-T20S structure indicated that the hydroxyl of ZYA's tyrosine side chain interacted with a water molecule that further hydrogen bonded with αL21 and αE25. To elucidate the relevance and importance of this water interaction, we mutated αE25 (*Chuah et al., 2023a*). αE25A proteasome mutants exhibited an upregulated basal activity that was not further activated by PAN or PA26, suggesting it plays an important role in gating. To elucidate whether a similar mechanism occurs in αV24Y T20S open gate conformation, we sought to identify where waters might be found within the

inter-subunit pocket. Broadly, we observed significantly fewer waters in the inter-subunit pockets of αV24Y T20S than we did in the WT T20S and ZYA-T20S. We note that αV24Y T20S's resolution is not as high as ZYA-T20S but is reasonable for predicting water coordinates. Because no common water was found in the two different open states, the necessity and role of specific water molecule interactions in gate-opening could not be elucidated from these structures.

Ultimately, the findings described indicate that the occupancy of the hydrophobic pocket within the α inter-subunit pocket is sufficient to induce gate-opening in archaeal proteasomes. The application of this finding in the mammalian system relies on further identification of the critical inter-subunit pockets that need to be occupied by the HbYX motif in that system to stabilize gate-opening. Our prior studies involving ZYA have indicated a minimum of two HbYX motifs binding for gate-opening in the mammalian system (*Chuah et al., 2023b*). However, the mammalian 20 S has 7 different pockets that can be occupied in different ways by 6 different Rpt1-6 C-termini, so the specific inter-subunit pockets that need C-termini/HbYX motifs bound to activate gate opening have not been worked out. All cryo-EM studies to date agree that occupancy of five Rpt C-termini is sufficient for gate opening, but which of those are required for gate opening is not known. Further progress on the HbYX mechanism based on the results thus far, would be to identify the specific inter-subunit pockets that need to be occupied by the HbYX motif to induce gate-opening in the eukaryotic proteasomes.

While this study provides significant insights, it is important to acknowledge certain limitations. A key limitation stems from using the homoheptameric archaeal T20S as our model. Although this simpler system allows for more reliable dissection of fundamental mechanisms, and core elements like HbYX-induced gate opening are conserved at the intersubunit pocket level, the overall T20S and eukaryotic 20 S/26 S proteasomes differ significantly in their complexity. Specifically, our engineered αV24Y mutation results in a tyrosine constitutively occupying all seven identical hydrophobic pockets. This contrasts with the eukaryotic proteasome, which possesses seven distinct α-subunit pockets that interact with various Rpt C-termini through dynamic binding. Moreover, the specific Rpt C-termini interactions—whether acting individually or cooperatively—that are essential to drive gate opening in the eukaryotic system remain incompletely understood. Therefore, while insights from our archaeal system are valuable for understanding general principles, direct comparisons and extrapolations to the intricate allostery and interaction complexities of the eukaryotic 26 S proteasome must be made with caution.

The work presented here is conducted with the intention of understanding proteasome activation mechanisms and developing the molecular framework for restoring or activating proteasome activity to combat the impairment consistently observed in neurodegenerative diseases, which is barred by our limited understanding of proteasome regulation. Here, we advance the field's understanding of proteasomal gate-opening by extending beyond the vague idea that HbYX motif binding induces gate opening to pinpointing a sole molecular interaction that significantly and sufficiently opens the proteasome's gate. Our findings inform proteasome-activating drug discovery efforts to focus on small molecules that occupy the HbYX hydrophobic pocket.

## Materials and methods
### Proteasome purifications
*T. acidophilum* wild-type 20 S, αV24Y 20 S, or αV24F 20 S proteasomes were similarly purified as described (*Dorn et al., 1999*), using the 8XHis tags on the C-terminus of the β subunits. All T20S mutants were generated by overlapping PCR site-directed mutagenesis.

### Proteasome activity assays
Fluorogenic substrate peptide, LFP (Mca-AKVYPYPME-Dpa(Dnp)-amide), was synthesized by EZBiolabs. LFP was dissolved in DMSO and incubated with proteasomes at 3 µM final concentration. The final concentration of DMSO in activity assays was 1%. Protein concentrations were determined by Bradford assay (Thermo Scientific). 0.25 µg archaeal proteasomes and LFP peptides were added to 25 µL Tris-based reaction buffer at 45 °C. Assays were performed for 30 min to an hour and analyzed using BioTek Gen5 Data Analysis software. Activity was measured as relative fluorescence units/ minute (rfu/min), generating a curve which was used to calculate the initial velocity, according to the

slope of the curves. 'Fold Activation' was calculated by dividing the average initial velocity of mutant proteasomes against the average initial velocity of wild-type proteasomes.

## Cryo-EM sample preparation and data collection

Copper Quantifoil R 1.2/1.3 300 mesh (EMS) grids were cleaned using a PELCO easiGlow Glow Discharge cleaning system. A volume of 3 µL of 0.75 mg/mL αV24Y-T20S or αV24F T20S (suspended in 50 mM Tris pH 7.4, 150 mM NaCl) sample was placed onto a grid, and then flash frozen in liquid ethane using a manual plunge freeze apparatus. Data collection was done using a Titan Krios transmission electron microscope (Thermo Fisher) operated at 300 kW and a magnification of x81,000, which resulted in 0.503 Å/px. Images were collected using a Falcon IIIEC direct electron detector camera equipped with a K3/GIF operating in counting and super-resolution modes. Electron dose per pixel of 50 e-/Å2 was saved as 40 frame movies within a target defocus range of –2.5 to –1.25. All the data was collected using cryoSPARC software (Structura Biotechnology Inc) (*Punjani et al., 2017*).

## Cryo-EM single particle analysis

Cryo-EM images of the αV24Y T20S and αV24F T20S proteasome were analyzed using cryoSPARC. Schematic for cryo-EM single-particle data processing available in the supplement. αV24Y T20S: From 3663 movies collected, we picked 396,542 particles after two rounds of 2D Classification to generate an Ab-initio model, which was used for homogeneous refinement (using D7 symmetry). Particle processing scheme and validations are shown in *Figure 1—figure supplement 1 & 2*.

αV24F T20S: From 4920 movies collected, we picked 542722 after two rounds of 2D Classification to generate an Ab-initio model, which was used for homogenous refinement (using D7 symmetry). Particle processing scheme and validations are shown in *Figure 7—figure supplement 1 & 2*.

All representations of the T20S proteasome complex were created using PyMOL 2.5.4 and UCSF ChimeraX v1.7.1 (*Goddard et al., 2018*; *Pettersen et al., 2021*).

## Confidence map generation

Small conformational changes (e.g. ~1 Å) with larger allosteric effects altering 20 S gating and enzymatic activity are common features in the 20 S proteasome, which has been published extensively. However, to more rigorously demonstrate that these changes are statistically significant at the map data level, we used CPP-EM to generate confidence maps of our WT and V24Y structures (*Figure 4—figure supplement 1B*). Confidence maps statistically interpret cryo-EM densities by testing each voxel against background noise and controlling the false discovery rate (FDR). We thresholded our maps of WT and V24Y at an FDR of 0.01 to highlight regions with statistically significant density. We then overlaid these maps in Chimera to show that they clearly show conformational changes even at an FDR of 0.01, especially at the specific regions the reviewer was concerned about.

## Atomic model building

The atomic models were built using PDB: 8F7K as a template, rigid body fitting into the electron density map using PHENIX 1.19.2-4158 (*Liebschner et al., 2019*; *Adams et al., 2010*; *Afonine et al., 2018*). The docked models were subjected to a cycle of morphing and simulated annealing, five real-space refinement macrocycles with atomic displacement parameters, secondary structure restraints, and local grid searches in PHENIX. Consequently, the models were refined by oscillating between manual real-space refinement in WinCoot (*Emsley et al., 2010*) 0.9.8.1 EL and real-space refinement in PHENIX (five macrocycles, without morphing and simulated annealing). Waters were added to models using PHENIX Douse. The final model of αV24Y T20S was deposited at wwPDB: 9BUZ.

## Statistical analysis

Data were analyzed in GraphPad or Excel using an unpaired Student's t-test (Prism). For all statistical analyses, a value of p<0.05 was considered significant.

## Acknowledgements

We thank the members of the Smith Lab for the helpful and valuable discussions, especially Giovanni Howells for his meticulous review of this manuscript. We also thank Thomas (Tom) C Terwilliger, PhD, at the Los Alamos National Laboratory for his gracious guidance and assistance in using Phenix.

Transmission electron micrographs were recorded at the University of Virginia Molecular Electron Microscopy Core facility (RRID:SCR_019031), which is supported in part by the School of Medicine and built with NIH grant G20-RR31199. In addition, the Titan Krios (S10-RR025067), Falcon II/3EC direct detector (S10-OD018149), and K3/GIF (U24-GM116790) were purchased in part or in full using the designated NIH grants. Molecular graphics and analyses performed with UCSF ChimeraX, developed by the Resource for Biocomputing, Visualization, and Informatics at the University of California, San Francisco, with support from National Institutes of Health R01-GM129325 and the Office of Cyber Infrastructure and Computational Biology, National Institute of Allergy and Infectious Diseases. This work was supported by National Institute of Health grant R01AG064188 (DMS).

## Additional information

### Funding

| Funder | Grant reference number | Author |
|---|---|---|
| National Institute on Aging | R01AG064188 | David M Smith |

The funders had no role in study design, data collection and interpretation, or the decision to submit the work for publication.

### Author contributions

Janelle JY Chuah, Formal analysis, Supervision, Validation, Investigation, Methodology, Writing – original draft, Writing – review and editing; Madalena R Daugherty, Investigation; David M Smith, Conceptualization, Resources, Data curation, Formal analysis, Supervision, Funding acquisition, Validation, Investigation, Methodology, Writing – original draft, Project administration, Writing – review and editing

### Author ORCIDs
Janelle JY Chuah ⓘ http://orcid.org/0000-0002-1139-694X
Madalena R Daugherty ⓘ http://orcid.org/0009-0004-9274-4134
David M Smith ⓘ https://orcid.org/0000-0002-1502-676X

Reviewer #1 (Public review): https://doi.org/10.7554/eLife.106611.3.sa1
Reviewer #2 (Public review): https://doi.org/10.7554/eLife.106611.3.sa2
Author response https://doi.org/10.7554/eLife.106611.3.sa3

## Additional files

### Supplementary files
MDAR checklist

### Data availability
Cryo-EM maps are deposited in the Electron Microscopy Data Bank (EMDB) under accession codes, EMD-44914 (αV24F T20S) and EMD-44926, coordinates are available from the RCSB Protein Data Bank under accession codes, 9BUZ (αV24Y T20S), 8F7K (ZYA-T20S), 8F6A (WT T20S).

The following datasets were generated:

| Author(s) | Year | Dataset title | Dataset URL | Database and Identifier |
|---|---|---|---|---|
| Chuah J, Smith D | 2025 | Thermoplasma acidophilum 20S proteasome - alphaV24Y | https://doi.org/10.2210/pdb9BUZ/pdb | Worldwide Protein Data Bank, 10.2210/pdb9BUZ/pdb |

*Continued on next page*

*Continued*

| Author(s) | Year | Dataset title | Dataset URL | Database and Identifier |
|---|---|---|---|---|
| Chuah J, Smith D | 2025 | Thermoplasma acidophilum 20S proteasome - alphaV24Y | https://www.ebi.ac.uk/emdb/EMD-44926 | EMDataBank, EMD-44926 |
| Chuah J, Smith D | 2025 | Thermoplasma acidophilum 20S proteasome alphaV24F | https://www.ebi.ac.uk/emdb/EMD-44914 | EMDataBank, EMD-44914 |

The following previously published datasets were used:

| Author(s) | Year | Dataset title | Dataset URL | Database and Identifier |
|---|---|---|---|---|
| Chuah J, Smith D | 2023 | Thermoplasma acidophilum 20S proteasome - wild type bound to ZYA | https://doi.org/10.2210/pdb8f7k/pdb | Worldwide Protein Data Bank, 10.2210/pdb8f7k/pdb |
| Chuah J, Smith D | 2023 | Thermoplasma acidophilum 20S proteasome - wild type | https://doi.org/10.2210/pdb8f6a/pdb | Worldwide Protein Data Bank, 10.2210/pdb8f6a/pdb |

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
