## [Editor Report · eLife Assessment]

This **valuable** manuscript describes cryo-EM structures of archaeal proteasomes that reveal insights into how occupancy of binding pockets on the 20S proteasome regulates proteasome gating. The evidence supporting these claims is **convincing**, although the extrapolation of these findings to the more complex eukaryotic proteasome may prove challenging. This work will be of high interest to researchers interested in proteasome structure and regulation.

---

## [Referee Report · Reviewer #1 (Public review)]

Summary:

In this manuscript, Chua, Daugherty, and Smith analyze a new set of archaeal 20S proteasomes obtained by cryo-EM that illustrate how the occupancy of the HbYX binding pocket induces gate opening. They do so primarily through a V24Y mutation in the α-subunit. These results are supported by a limited set of mutations in K66 in the α subunit, bringing new emphasis to this unit.

Strengths:

The new structure's analysis is comprehensive, occupying the entire manuscript. As such, the scope of this manuscript is very narrow, but the strength of the data are solid, and they offer an interesting and important new piece to the gate-opening literature.

Weaknesses:

Extrapolating from the core HbYX activating motif shared by Archae and Eukaryotes to the specific operations of gate opening, which is more elaborate in eukaryotes, may prove challenging.

---

## [Referee Report · Reviewer #2 (Public review)]

Summary:

The manuscript by Chuah et al. reports the experimental results that suggest the occupancy of the HbYX pockets suffices for proteasome gate opening. The authors conducted cryo-EM reconstructions of two mutant archaeal proteasomes. The work is technically sound and may be of special interest in the field of structural biology of the proteasomes.

Strengths:

Overall, the work incrementally deepens our understanding of the proteasome activation and expands the structural foundation for therapeutic intervention of proteasome function. The evidence presented appears to be well aligned with the existing literature, which adds confidence in the presentation.

Comments on revisions:

The authors have addressed all my questions.

---

## [Author Response]

The following is the authors’ response to the original reviews

**Public Reviews:**

**Reviewer #1 (Public review):**
Summary:In this manuscript, Chua, Daugherty, and Smith analyze a new set of archaeal 20S proteasomes obtained by cryo-EM that illustrate how the occupancy of the HbYX binding pocket induces gate opening. They do so primarily through a V24Y mutation in the αsubunit. These results are supported by a limited set of mutations in K66 in the α subunit, bringing new emphasis to this unit.Strengths:The new structure's analysis is comprehensive, occupying the entire manuscript. As such, the scope of this manuscript is very narrow, but the strength of the data is solid, and they offer an interesting and important new piece to the gate-opening literature.Weaknesses:Major Concerns(1) This manuscript rests on one new cryo-EM structure, leading to a single (albeit convincing) experiment demonstrating the importance of occupying the pocket and moving K66. Could a corresponding bulky mutation at K66 not activate the 20S proteasome?

Thank you for this insightful question. We believe such a mutation would likely not activate the proteasome, and would likely be detrimental to gate opening. Our previous work (Smith et al., Molecular Cell, 2007), and data presented in this manuscript, demonstrate that a K66A mutation, which removes the side chain, blocks 20S gate opening. Furthermore, our new αV24Y T20S structure reveals that Lys66 forms specific hydrogen bonds with surrounding residues that are crucial for stabilizing the open gate conformation (Fig. 5). An aromatic or bulky hydrophobic mutation at this position would be unable to form these essential hydrogen bonds and would likely disrupt the necessary stabilizing interactions.

(2) To emphasize the importance of this work, the authors highlight the importance of gateopening to human 20S proteasomes. However, the key distinctions between these proteasomes are not given sufficient weight.(a) As the authors note, the six distinct Rpt C-termini can occupy seven different pickets. However, how these differences would impact activation is not thoroughly discussed.

We appreciate the reviewer's point regarding the complexities of eukaryotic 26S proteasome activation. While our manuscript discusses some aspects of this, we agree that a detailed mechanistic extrapolation from our archaeal T20S model to the diverse interactions within the human 26S proteasome is challenging. As we elaborate in our response to Reviewer #2 (Recommendation #3), the significant differences in α-ring composition (homoheptameric vs. heteroheptameric) and the multifactorial nature of Rpt C-termini binding make direct, wide-reaching speculations about specific pocket contributions in the eukaryotic system difficult at this stage. Our aim was to focus on the conserved fundamental role of the HbYX hydrophobic pocket itself.

(b) With those other sites, the relative importance of various pockets, such as the one controlling the α3 N-terminus, should be discussed more thoroughly as a potential critical difference.

The reviewer raises an excellent point about the regulation of specific α-subunits, like the α3 N-terminus, which acts as a lynchpin in gating. Understanding its precise regulation in the eukaryotic 26S proteasome is indeed a key goal in the field. However, determining which specific HbYX binding events (e.g., in the α2-α3 pocket, the α3-α4 pocket, or cooperative binding across multiple pockets) control the α3 subunit's conformation is beyond the scope of what our current T20S structural data can definitively inform. The cooperative nature of HbYX binding and its precise allosteric consequences across the heteroheptameric α-ring are complex questions that remain to be fully elucidated in the eukaryotic system. Our study focuses on demonstrating the sufficiency of hydrophobic pocket occupancy for activation in a conserved manner, which we propose is a fundamental aspect of HbYX action. Identifying which of the seven distinct eukaryotic hydrophobic pockets must be engaged for full activation remains an important area for future research.

(c) These differences can lead to eukaryote 20S gates shifting between closed and open and having a partially opened state. This becomes relevant if the goal is to lead to an activated 20S. It would have been interesting to have archaea 20S with a mix of WT and V24Y α-subunits. However, one might imagine the subclassification problem would be challenging and require an extraordinary number of particles.

We agree with the reviewer that exploring mixed subunit populations is an interesting idea, particularly given the dynamic and potentially partially open states of eukaryotic proteasomes. We have previously considered co-expressing WT and V24Y α-subunits. However, the interpretation of such experiments would be challenging. With 14 potential sites for mutant incorporation across the two homoheptameric α-rings, a heterogeneous population of proteasomes with varying numbers and arrangements of V24Y subunits would be generated. Correlating any observed changes in activity or structure (e.g. via cryoEM subclassification, would be exceedingly difficult) to specific stoichiometries or arrangements of mutant subunits would be highly complex and likely inconclusive for deriving clear mechanistic insights.

(d) Furthermore, the conservation of the amino acids around the binding pocket was not addressed. This seems particularly important in the relative contribution of a residue analogous to K66 or V24.

We apologize for the mislabeled figure title in the previous submission, which may have made this information less accessible. We have now corrected the title for Supplemental Figure S10 (previously S9). This figure presents the sequence alignment showing the conservation of residues in and around the HbYX hydrophobic pocket, including those analogous to T20S αV24, αL21, and αA154. As discussed in the manuscript, key residues that form this pocket, such as those corresponding to and surrounding T20S L21 and A154, are indeed well conserved in human α-subunits. This conservation supports the relevance of our findings to eukaryotic proteasomes.

**Reviewer #2 (Public review):**
Summary:The manuscript by Chuah et al. reports the experimental results that suggest the occupancy of the HbYX pockets suffices for proteasome gate opening. The authors conducted cryo-EM reconstructions of two mutant archaeal proteasomes. The work is technically sound and may be of special interest in the field of structural biology of the proteasomes.Strengths:Overall, the work incrementally deepens our understanding of the proteasome activation and expands the structural foundation for therapeutic intervention of proteasome function. The evidence presented appears to be well aligned with the existing literature, which adds confidence in the presentation.Weaknesses:The paper may benefit from some minor revision by making improvements on the figures and necessary quantitative comparative studies.

We appreciate the reviewers thoughtful critique of our manuscript and have made the requested changes and provided further perspectives mentioned below.

**Recommendations for the authors:**

**Reviewer #1 (Recommendations for the authors):**
(1) Line 467: Mammalian should be replaced with eukaryotic.

Done.

(2) Figure 1 Caption: The descriptions of the blue and green boxes should be described in panel A's caption rather than waiting until panel C.

Done.

(3) Figure 2 A: For greater clarity, the asterisks should be replaced with the numbers H4, H5, and H6.

Done.

(4) Figure 7 caption: The panels are misannotated. What is listed as E should become D, and what is listed as F should become E.

Done.

(5) The title for Figure S9, "αV24Y T20S validation," is inappropriate. A better title should discuss the sequence conservation of those amino acids. Why is the arrow drawing attention to L21 when the paper is about V24? There should be a corresponding alignment that includes K66.

Thank you for pointing out the title issue for Figure S10 (previously S9) this has now been corrected to reflect its focus on sequence conservation. The arrow highlighting L21 (and its eukaryotic analogues) is intended to draw attention to a key residue that, along with A154, forms part of the hydrophobic pocket occupied by V24Y. As detailed in the main text and shown in Figures 3C, 3D, and 4G, measurements involving L21 were used to demonstrate the widening of this pocket upon V24Y mutation or ZYA binding.

**Reviewer #2 (Recommendations for the authors):**
The authors might consider improving the manuscript by addressing the following minor issues:(1) Figure 1: it might be easier for readers to understand what the authors meant to show by superimposing the atomic model of the mutated sidechain with the density map. In this case, the density map could be rendered half-transparent, or it could be represented by mesh.

We appreciate this suggestion for enhancing Figure 1. While we agree that showing the model fit within the density is valuable, we found that incorporating this directly into the comparative overlay panels of Figure 1 (which already depict multiple aligned density maps) made the figure overly complex and visually detracted from its primary message of comparing overall conformational states. However, we do provide a clear illustration of the model-to-map fit for the αV24Y T20S structure in Supplemental Figure S3, where the atomic model is shown within the transparent map surface. Furthermore, all our maps and models are publicly available, and we encourage interested readers to perform detailed comparisons. We believe this approach balances clarity in the main figure with the provision of detailed validation data.

(2) What is the solvent-inaccessible surface area of the mutated side-chain buried by its hydrophobic interaction with the HbYX pockets? How is this buried surface area compared to the solvent-accessible surface area of the HbYX pocket without the mutation?

We appreciate the idea of another visual to answer the question and provide the reader with a better perception of this pocket in the WT versus V24Y T20S. To address this we added a new Supplemental Figure 7 with surfaces showing this comparison including each separate pocket and an overlay with solid and mesh surfaces. We also added this line to the text: “Moreover, molecular surface representations of the hydrophobic pocket clearly show occupancy by the mutant tyrosine’s side chain (Fig. S7)”.

(3) Based on the data of the buried surface area of the mutated side-chain (requested above), can the authors make some quantitative comparison with the activated eukaryotic proteasome (either human or yeast 26S) with the alpha-pocket occupied with HbYX motifs from Rpt subunits? How similar are they?

This is a thoughtful suggestion, and we understand the interest in directly comparing pocket occupancy across systems. While we draw general parallels regarding HbYXdependent activation in the discussion, we believe a direct quantitative extrapolation of specific surface area occupancies from our T20S V24Y mutant to the eukaryotic system would be overly speculative and unlikely to yield further definitive insights into the eukaryotic gate-opening mechanism at this time. The primary reason for this is the significant disparity in complexity between the archaeal T20S and eukaryotic 26S proteasomes. The eukaryotic α-ring is a heteroheptamer, composed of seven distinct αsubunits, which creates seven non-identical inter-subunit pockets. In contrast, our study utilizes the homoheptameric archaeal T20S. Furthermore, eukaryotic 26S proteasome activation involves the intricate binding of multiple C-terminal tails from the six different Rpt ATPase subunits of the 19S regulatory particle. These C-termini include various HbYX motifs as well as non-HbYX tails, and they interact with the diverse α-subunit pockets in a highly complex, multifactorial manner that drives what appears to be an allosteric mechanism for gate regulation.

Crucially, the precise number of C-termini required for 20S gate-opening in the eukaryotic system, the specific combination of these Rpt C-termini, and even the exact inter-subunit pockets that must be occupied to induce robust gate opening are still areas of active investigation and are not resolved (as discussed in our manuscript). Therefore, attempting to extrapolate nuances, such as the precise degree of hydrophobic pocket occupancy from our single, engineered αV24Y side-chain (which models one specific type of Hb-pocket interaction in a simplified system) to each of the potentially five or more different Rpt Ctermini interactions within the various 20S inter-subunit pockets in the eukaryotic 26S proteasome, would involve too many assumptions and would not provide reliable predictive power to understand mechanism.

However, regarding the fundamental question of how a hydrophobic group occupies the HbYX pocket in our archaeal model system, we believe Figure 4D provides relevant insight that may address the reviewer's underlying curiosity. This figure carefully illustrates the spatial overlap, showing that the engineered αV24Y side-chain and the hydrophobic 'Z' group of the ZYA HbYX-mimetic occupy the same region within the T20S inter-subunit hydrophobic pocket. This provides a clear visual comparison of this key 'Hb' interaction in our defined and structurally characterized system.

(4) It may be helpful that at the end of the discussion, the authors make some comments on how the current results might offer insights into the eukaryotic proteasome activation, and on what the limitations of the current study are.

We thank the reviewer for this suggestion. We agree that discussing the implications for eukaryotic proteasome activation and the study's limitations is important.

Insights into Eukaryotic Proteasome Activation:

We have indeed discussed how our current findings with the αV24Y T20S mutant offer insights into eukaryotic proteasome activation in the Discussion section. To briefly summarize:

(1) Conservation of the Target Site: Our study highlights that the key residues forming the hydrophobic pocket targeted by the αV24Y mutation (αL21 and αA154 in T20S) are well-conserved in the human 20S α-subunits (as shown in Fig. S9). This suggests that the mechanism of inducing gate opening through occupancy of this specific hydrophobic 'Hb' pocket by an aromatic residue is a plausible strategy for activating eukaryotic proteasomes.

(2) Relevance of the IT Switch: The αV24Y mutation, by occupying the Hb-pocket, allosterically affects the conserved IT switch, promoting an open-gate conformation. As detailed in our previous work (Chuah et al., Commun. Biol. 2023; Ref. 31 in the current manuscript), this IT switch mechanism is also functionally conserved in most human α-subunits. The current study reinforces that direct manipulation of the Hb-pocket is sufficient to trigger this conserved downstream gating machinery.

(3) Therapeutic Implications: These findings further pinpoint the HbYX hydrophobic pocket as a specific and promising target for the design of small molecule proteasome activators aimed at human proteasomes.

While these parallels are informative, we reiterate our caution as also mentioned in response to comment #3 and in the manuscript regarding direct quantitative extrapolation due to the increased complexity of the heteroheptameric eukaryotic α-ring and the multifactorial nature of Rpt C-termini interactions.

We also agree that we should add a statement regarding key limitation raised by the reviewer, to our manuscript. Below is the key limitations paragraph that has been added to the penultimate paragraph of the discussion:

“While this study provides significant insights, it is important to acknowledge certain limitations. A key limitation stems from using the homoheptameric archaeal T20S as our model. Although this simpler system allows for more reliable dissection of fundamental mechanisms, and core elements like HbYX-induced gate opening are conserved at the intersubunit pocket level, the overall T20S and eukaryotic 20S/26S proteasomes differ significantly in their complexity. Specifically, our engineered αV24Y mutation results in a tyrosine constitutively occupying all seven identical hydrophobic pockets. This contrasts with the eukaryotic proteasome, which possesses seven distinct α-subunit pockets that interact with various Rpt C-termini through dynamic binding. Moreover, the specific Rpt Ctermini interactions—whether acting individually or cooperatively—that are essential to drive gate opening in the eukaryotic system remain incompletely understood. Therefore, while insights from our archaeal system are valuable for understanding general principles, direct comparisons and extrapolations to the intricate allostery and interaction complexities of the eukaryotic 26S proteasome must be made with caution.”